# Biodegradation Properties of Cellulose Fibers and PLA Biopolymer

**DOI:** 10.3390/polym15173532

**Published:** 2023-08-24

**Authors:** Ružica Brunšek, Dragana Kopitar, Ivana Schwarz, Paula Marasović

**Affiliations:** 1Department of Materials, Fibres and Textile Testing, Faculty of Textile Technology, The University of Zagreb, Prilaz baruna Filipovića 28a, 10000 Zagreb, Croatia; 2Department of Textile Design and Management, Faculty of Textile Technology, The University of Zagreb, Prilaz baruna Filipovića 28a, 10000 Zagreb, Croatia; dragana.kopitar@ttf.unizg.hr (D.K.); ivana.schwarz@ttf.unizg.hr (I.S.); paula.marasovic@ttf.unizg.hr (P.M.)

**Keywords:** biodegradation, hemp, jute, sisal, viscose, PLA biopolymer, soil burial test, properties, soil quality

## Abstract

This paper investigates the biodegradation properties of cellulose fibers and PLA biopolymer. For that purpose, hemp, jute, and sisal fibers as lignocellulose fibers; viscose fibers (CV) as regenerated cellulose; and polylactide (PLA) as biopolymer were buried in farmland soil for periods of 2, 4, 7, 9 and 11 days under controlled conditions. The influence of their biodegradation on the fiber mechanical properties, bacteria and fungi population, as well as on the soil quality were investigated. After exposure to microorganisms, analyses of the fibers’ morphological (SEM), chemical (FTIR), and thermal (TGA) properties were conducted to achieve a comprehensive understanding of their biodegradability. The analysis concluded that lignin and pectin content have a greater impact on the biodegradation of hemp, jute, and sisal fibers than factors like crystallinity and degree of polymerization. The viscose fibers showed lower biodegradability despite their lower degree of polymerization, indicating a resistance to biodegradation due to the “skin” formed during the spinning process. PLA fibers experienced chemical hydrolysis and significant microbial attack, resulting in reduced tenacity. The acquired findings yield valuable insights into the biodegradability of the fibers, thereby facilitating the selection of appropriate fibers for the development of environmentally sustainable products. Notably, a literature review revealed a paucity of research on fiber biodegradability, underscoring the significance of the present study’s contributions.

## 1. Introduction

Biodegradation is a chemical reaction that causes a significant change in the material structure characterized by fragmentation and decreasing properties (molecular weight, tenacity, etc.), where eventually, a material degrades into substances such as water, carbon dioxide, methane, biomass, and humic matter [1,2,3]. Biodegradation can occur through abiotic or biotic processes. The degradation of polymers involves four major abiotic mechanisms: mechanical, thermal (or thermo-oxidative), photo (or photo-oxidative), and hydrolytic (or chemical) degradation. These mechanisms contribute to the reduction in strength and fragmentation of polymeric molecules, ultimately leading to the formation of smaller units that are biodegradable. Biotic material biodegradation occurs through the action of microorganisms by enzymatic action in living organisms, such as bacteria, protozoa, algae, and fungi, that require a carbon source for growth and reproduction [1,4].

The microbial degradation of polymers involves two fundamental steps: depolymerization or chain cleavage, followed by mineralization. During the initial step, the lengthy polymer chain is transformed into smaller oligomer fragments that can be assimilated. This process may be facilitated by hydrolysis and/or oxidation and extracellular enzymes. In the subsequent step, which takes place within the microbial cell, the smaller fragments are converted into biomass, minerals, salts, water, and various gases such as CO_2_, CH_4_, N_2_, and H_2_. Metabolic energy is typically derived by the cell through the mineralization process. The biodegradation process can exhibit several variations depending on factors such as the specific polymer, organisms involved, and environmental conditions. Nonetheless, enzymes play a crucial role at some stage of the process. Enzymes, being biological catalysts, have the ability to significantly enhance reaction rates (108–1020 fold) even in environments where chemical reactions would otherwise be unfavorable [5].

In order for to microorganisms survive, certain conditions such as those provided by light, water, and oxygen are needed. In nature, different materials biodegrade at different periods. According to recent studies, temperature is a significant factor influencing the biodegradation process (especially in biopolymers) since the reproduction of microorganisms is faster in warmer conditions [6,7,8]. When a material is buried, microorganisms gradually degrade the buried sample over a certain period, resulting in mass loss and a decrease in mechanical properties, oxygen consumption (O_2_), and carbon dioxide production (CO_2_). The biodegradation process is indicated by the percentage of mass loss, decrease in mechanical properties, and the amount of produced CO_2_. The determination of material mass loss by the gravimetric method is not a reliable indicator of biodegradability. The migration of water-soluble additives, the dissolution of water-soluble polymers, as well as melting can cause a mass loss in the test sample. Any conclusion about biodegradability based on mass loss should be supported by other standardized methods for material biodegradation (e.g., tensile tests, respirometric tests, etc.) [9,10,11].

Lignocellulose fibers are constituted by three structural polymers (the polysaccharides cellulose, hemicelluloses, and the aromatic polymer lignin) as well as by some minor non-structural components (i.e., proteins, extractives, and minerals) [12,13]. Cellulose is the basic building material of the lignocellulose fiber, with a small proportion of hemicellulose and so-called non-cellulosic components: lignin, pectin, fats and waxes, water, pigments, minerals, and ashes in different percentages. The chemical composition of lignocellulose fibers depends on sources, climatic conditions, soil quality, the retting process, and the applied method of determining the chemical composition [14,15,16]. In addition, lignocellulose fibers also differ in crystallinity, degree of polymerization, and method of fiber production. They are considered environmentally friendly fibers known to degrade very easily by microorganisms when buried. However, not all cellulose fibers are characterized by the same biodegradability due to mutual differences in chemical structure and physical characteristics [16,17,18]. 

In contrast to lignocellulose fibers, viscose consists of 100% cellulose without any other non-cellulosic components. In addition, viscose fibers are hygroscopic and have a high absorption of moisture from the environment. Due to the low degree of polymerization, weaker orientation, and lower proportion of crystallinity, in the wet state, these fibers swell and characteristically stiffen and deform. Therefore, the highest biodegradability is expected regardless of its low crystallinity [19,20].

Two main groups of microorganisms are responsible for the enzymatic degradation of cellulose, namely bacteria and fungi. Biodegradation in the presence of microorganisms takes place from the surface inwards, where after breaking the cuticle, organisms penetrate through the secondary wall into a lumen where they grow [21,22]. The cellulose fiber biodegradation is directly related to its degree of crystallinity; first, the amorphous regions are decomposed and then the crystalline ones. Therefore, the fibers with a lower degree of crystallinity will have a greater biodegradation degree [21,23]. 

PLA (polylactic acid polymer) is a biodegradable thermoplastic polymer produced from natural and renewable sources, such as corn and sugar, beet, and starch, which is characterized by environmental benefits and good mechanical properties [24]. Like other polymers, the PLA biopolymer has its disadvantages. Recently, intensive research and modification by physical or chemical methods have been carried out to overcome the disadvantages of PLA biopolymers with the aim of synthesizing polymers with good biodegradability, flexibility, and heat resistance. PLA biodegradation is influenced by polymer characteristics such as degree of crystallinity, molecular weight, morphological structure, molecular and supramolecular structures, etc. In addition, the biodegradation of PLA is influenced by environmental conditions such as the presence of water and moisture, temperature, acidity conditions, oxygen content, and the type and activity of microorganisms. Increasing the amount of moisture increases the rate of hydrolysis of PLA and the development and reproduction of microorganisms. As a result, PLA biodegradation will be faster in humid surroundings than in dry ones. Another influencing factor is temperature. The rate of hydrolysis and microbial activity increase with temperature, but the activity of microorganisms can be severely decreased or completely stopped at excessively high temperatures. Exposure to UV radiation is another environmental element that influences PLA degradation. UV radiation decreases PLA’s physical integrity and increases its brittleness and breaking stress [24]. The degradation of PLA in the biological environment is divided into two steps: the non-enzymatic melting of ester groups and the random degradation of polymer chains of low molecular weight by microbes to produce carbon dioxide and water [25].

In recent years, the waste disposal problem, aggravated by the increased use of disposable materials resistant to biodegradation, has increased interest in the biodegradability of materials. A lot of research has been carried out on the biodegradation of textile fabrics (woven, nonwoven, and knitted) and composites produced from cellulose fibers, PLA fibers, and blends. Considering that textile fabrics’ biodegradation is affected by their structure (thickness, production process, weave, yarn, etc.), systematic research on fiber biodegradation is essential. The systematic review of the available literature on fibers’ and polymers’ biodegradability elucidated that the hydrophilic cellulose polymer easily biodegrades by the action of microorganisms [8,26,27,28]. Therefore, microorganisms secrete the enzyme cellulase that breaks the cellulose chain into cellobiose units that break down further into glucose or glucose derivatives. Modifying cellulose polymer by blocking hydrophilic hydroxyl groups can increase the resistance of the polymer to microbial attack [29]. Research on PLA polymers in the form of films with various thicknesses and fibers (materials were made out of Nature Works PLA pellets) showed that the thickness and material form (surface area of material exposed to soil, film, or fibers; single specimens or material mass) play a crucial role in the biodegradation process under soil burial conditions. Laboratory simulated soil burial results were found to be consistent with results obtained under field conditions [30]. The study on cotton, wool, and PLA fiber biodegradability in farmland soil conducted by soil burial test showed that wool and cotton samples were significantly degraded, while the PLA fiber showed a slight initial degradation. Although PLA fiber is considered degradable, the biodegradation of PLA in a natural environment proceeds relatively slowly compared to natural fibers. The study showed a more significant decrease in the tenacity than the fiber mass, probably due to the degrees of polymerization and crystallization by the microbe [31,32,33].

In previous research [34], the biodegradation properties of PLA, hemp, jute, and viscose for agro-textile non-wovens production were investigated using a soil burial test with humus soil. The study was conducted under controlled conditions in the laboratory and also under real weather conditions. The results showed that faster biodegradation occurred in fibers exposed to microorganisms under controlled conditions. This could be attributed to the more favorable conditions for microbial activities. The findings from fibers exposed to microorganisms under real weather conditions are also crucial since agro-textile materials are used in such conditions. Based on the results obtained, the investigation proceeded further.

Considering that textile fabrics’ biodegradation is affected by their structure and other characteristics such as thickness, production process, weave, yarn, etc., a systematic research on fiber biodegradation is needed. Therefore, the aim of this investigation is to define the biodegradation properties of hemp, jute, and sisal fibers as lignocellulose fibers; viscose fibers (CV) as regenerated cellulose; and polylactide (PLA) as biopolymer in farmland soil by a soil burial test. A comprehensive understanding of fiber biodegradation was achieved by analyzing the fibers’ mechanical properties, bacteria and fungi population, soil quality, and the morphological (SEM), chemical (FTIR), and thermal (TGA) properties following their exposure to microorganisms.

## 2. Methodology

The fibers investigated for determining the biodegradation properties included hemp (58.54 dtex), jute (31.02 dtex), sisal (48.06 dtex), and viscose (1.78 dtex) supplied by Derotex NA and PLA from cornstarch (6.84 dtex) supplied by NatureWorks BV, Plymouth, Minnesota, USA. The displayed units of fiber fineness (dtex) define 1 g of fiber per 10.000 m. These fibers were buried for five different periods of 2, 4, 7, 9, and 11 days and subsequently retrieved from the soil for testing purposes. The control sample of each fiber represents an unburied fiber (the buried period is equal to 0), which was used to compare the changes in investigated properties and assess their biodegradation. 

The fiber biodegradation was determined using a soil burial test according to ISO 11721-1:2001 [35] and ISO 11721-2:2003 [36] standards. Considering that there are currently no standards for fiber biodegradability testing, the standards were modified for the investigated fibers: the shape of fiber bundles was prepared to resemble as much as possible a flat textile; washing duration was extended to account for any residual soil present within the fiber bundle. To observe the progressive changes in fiber structure and decomposition, the fibers were buried in farmland soil for durations of 2, 4, 7, 9, and 11 days. For the optimal activity of microorganisms, soil moisture was 60 ± 5%, and soil pH was 6.8. To prevent fibers from mixing or overlapping, samples of each fiber type were buried in separate containers of unglazed pottery at a depth of 150 mm. Pots with fibers were placed in an incubator with a constant air humidity of 95 ± 5% and a temperature of 29 ± 1 °C. The fibers were removed from the soil after a defined period (2, 4, 7, 9, and 11 days), cleaned and washed in an ethanol/water solution (70/30 vol.%) for approximately 40 min, and afterwards dried at room temperature. The samples were then subjected to relevant tests to determine their biodegradation.

### 2.1. Mechanical Properties of the Fibers

The mass loss of fibers subjected to burial for a period of 2, 4, 7, 9, and 11 days was determined after removing moisture from the fibers (by heating the fibers for 24 h at 105 °C), cooling in a desiccator (for 1 h), and weighing. Mass loss was calculated as the percentage difference in mass before and after the burial test. 

Lignocellulose fibers are characteristic of a high level of non-uniform morphology that can be neither substantially affected nor removed by the production technology, but it noticeably affects fiber properties. Therefore, fiber tenacity was determined according to the standardized method (ISO 5079:2020 [37]) on Vibrodyn 400 (Lenzing, Austria) modified for fiber testing. Cogged steel clamps were placed on standard clamps, and the test was carried out at a speed of 3 mm/min. The selected measurement length was 5 mm to ensure that all elementary fibers in the test sample were captured during the tenacity tests. The number of measurements was increased from 50 to 100 per sample. The tenacity of the viscose and PLA fibers were determined according to the above standard without any modification of the test; i.e., the distance between the standard clamps was 20 mm with a speed of 100 mm/min, and the number of measurements was 50. Measurements of tested properties were performed on conditioned fibers.

### 2.2. Farmland Soil Quality

The total organic carbon (TOC) of the soil samples of buried fibers was determined according to the ISO 10694:1995 [38] standard. The accuracy of the analyses was controlled by RM, ISE 869, ISE 989, and ISE 851, ISE 920 Wepal (GAP compliant, 10% per sequence), with relative measurement error < 3%. The results are expressed as a percentage in dry matter (dried at 105 °C to constant mass). The total number of fungi and bacteria in the soil by was tested classical microbiological methods. The percentage of dry matter for all soil samples was determined after drying the samples. Soil samples in a sterile saline solution were homogenized for 5 min. After that, a series of dilutions were made, followed by the smear of relevant sample volume on solid nutrient media. Nutrient agar (NA) was used to determine the total number of bacteria, while Czapek agar was used to determine the total number of fungi. All samples were tested in triplicate. After the cultures had grown on nutrient media, the colonies were counted, and the CFU value of individual groups of microorganisms per gram of soil was determined. Chemical analysis of soil quality included the determination of pH in 1:5 suspension of soil in water (pH in H_2_O), in 1 mol/L potassium chloride solution (pH in KCl), or in 0.01 mol/L calcium chloride solution (pH in CaCl_2_), according to ISO 10390:2021 [39]; humus by sulfochromic oxidation method; organic carbon (C) by bicormat method; total nitrogen (N) by the Kjeldahl method; phosphorus (P_2_O_5_); and potassium (K_2_O) by Al method.

### 2.3. Morphological Analysis

Samples of both control and buried fibers were subjected to analysis using a scanning electron microscope (FE-SEM//Mira, Tescan, Brno, Czechia) at a magnification of ×1000. SEM images were obtained with an accelerating voltage of 5 kV. Before the SEM investigation, the samples underwent a chrome steaming process for 180 s to enhance their electrical conductivity.

### 2.4. FTIR Analysis

The FTIR spectra of both control and buried fiber samples were acquired using a Fourier-Transform Infrared Spectrometer (Perkin Elmer Spectrum 100 FT-IR, Waltham, MA, USA). The analyses were conducted at room temperature and under ambient humidity conditions. The solid samples in their original state were positioned on the ATR crystal, ensuring complete coverage and applying pressure. Spectra were recorded within the range of 4000 cm^−1^ to 380 cm^−1^, utilizing a resolution of 4 cm^−1^. Each spectrum was generated by averaging four individual scans.

### 2.5. Thermalgravimetric Analysis (TGA)

Thermalgravimetric analysis of the control and buried fiber samples was conducted using a Perkin Elmer Pyris 1 thermogravimetric analyzer (Perkin Elmer, Waltham, MA, USA). Before testing, the fibers were crushed into smaller fragments, with weights ranging from 7 mg to 10 mg. The samples were subjected to heating from 30 °C to 800 °C at a heating rate of 10 °C/min, under a nitrogen flow of 30 mL/min.

## 3. Results and Discussion

### 3.1. Fiber Mass Loss

A fiber biodegradation analysis includes assessing the fiber mass loss after burial in the soil. The fiber’s mass loss after burial in the soil for a buried period is expressed in percentages compared to the control samples and shown in Figure 1. 

The results of mass loss are in accordance with expectations that the mass loss of samples increases with the increasing time of exposure to microorganisms. The percentage mass loss in all investigated fibers was in a linear relationship with the progression of fiber biodegradation. Mass losses for all cellulosic fibers are higher regarding PLA biopolymer fibers due to their better ability to absorb and retain moisture, thereby decreasing biodegradation time. In addition, cellulosic fibers are highly sensitive to hydrolytic degradation by microorganisms such as fungi and bacteria from the soil. The breakage and degradation of the samples facilitate the colonization of the fibers by microorganisms. Complex polymer chains are enzymatically broken down by microorganisms into simpler organic molecules, which serve as a source of nourishment for the microorganisms, facilitating their multiplication. During the process of biodegradation, the molecular weight of the polymer decreases, and at the end, the individual polymer chains are completely biologically degraded into biomass (humus) and biogas (methane and carbon dioxide) [5].

The greatest mass loss after the longest period of exposure to microorganisms (11 days) was evident for hemp fibers, followed by jute and sisal fibers. The mass decrease for hemp fiber was 44.51%, while for sisal, it was 7.92%. It can be assumed that the different chemical compositions influence the biodegradation properties of tested fibers. Therefore, a higher proportion of lignin and pectin in sisal fibers prolonged the time of the fiber surface breakage and the penetration of microorganisms into the fiber structure. The researchers revealed that viscose fibers, which contain more amorphous and less crystalline area and thus have better hydrophilicity than most natural cellulose fibers, have the highest biodegradability. This study showed that the mass loss of viscose fibers was lower than the mass loss of hemp and jute fibers, i.e., 11.64%. As expected, the smallest mass decrease regarding the control sample was evident with PLA fibers. After 11 days of burial and exposure of PLA fibers to microorganisms, the mass loss was only 3.99%. 

### 3.2. Fibee Tensile Properties

The content of cellulose in lignocellulose fibers, the microfibrillar angle, and the degree of cellulose polymerization define different mechanical properties. The cellulose content influences their mechanical properties, where increasing the cellulose content increases fiber tenacity [20,40]. The second main component of cellulosic fibers is lignin, which creates a protective layer preventing the internal structure of fibers from degrading. Lignin is durable and not soluble in water, acting as a glue to cellulose and hemicellulose. All these differences influence the degradation behavior of cellulose fibers, resulting in different biodegradability, which was confirmed by the obtained tenacity results (Table 1). Since individual measurements are concerned, a relatively high coefficient of variation of lignocellulose fiber (ranging from 28.30% to 47.40%) was found, which is a consequence of the high non-uniformity of the morphology of lignocellulose fibers. For a clear interpretation of the biodegradation properties, the results are also shown in Figure 2, expressed as a percentage compared to control samples. 

After exposure of the fibers to soil microorganisms, the tenacity decreased as expected. The hemp fibers showed the highest tenacity decrease (66.17%) and thus biodegradability, which could be explained by the largest amount of cellulose (approx. 70%) and a small amount of lignin (approx. 6%). After the longest period of exposure to microorganisms (11 days), the jute fibers showed less tenacity decrease (24%), where jute fibers comprise 55–65% cellulose and around 12% lignin, which protects the fiber from external influences [22,23]. The tenacity decrease in sisal fibers, which contain an even higher amount of lignin (11–19%) than jute fibers, was only 17.58%. 

Unlike lignocellulose fibers, viscose fibers consist of 100% cellulose without other non-cellulosic components. Viscose fibers are hygroscopic and have a high moisture absorption from the environment. Due to the low degree of polymerization, weaker orientation and smaller proportion of crystallinity, and the highest tenacity decrease, respectively, biodegradability was expected. The observed decrease in the tenacity of viscose fibers after being exposed to microorganisms for the longest duration (11 days) amounted to a reduction of only 22.79%. This result indicates that the presence of a protective “skin” formed on the outer surface of the fibers during the spinning process, acting as a barrier, hampers the infiltration of microorganisms, thereby retarding the process of biodegradation [20,23]. During the spinning process, the outer layer of the fiber undergoes coagulation in the sedimentation bath, leading to the formation of a distinct “skin” on the fiber’s outer surface. The key distinction between the outer and inner parts of the fiber lies in the more densely distributed crystallites and higher resistance found in the skin, while the inner part remains partially amorphous. This difference in structure results in the outer layer, i.e., skin, slowing down the penetration of microorganisms, thereby impeding the biodegradation process.

Since the PLA polymer is largely resistant to attack by soil microorganisms, PLA degradation upon disposal in the environment is challenging. PLA will degrade quickly and disintegrate within weeks to months under high-temperature and high-humidity conditions [41]. The PLA fibers, which are not protected by the structural parameters of textile fabrics (woven, knitted, and non-woven) and in favorable conditions for the microorganisms activation, showed an unexpectedly large decrease in tenacity (37.23%).

Assessing the biodegradability rate of fibers presents significant challenges for several reasons. Although the gravimetric method is commonly used to evaluate biodegradation by measuring mass loss, it has limitations in accurately indicating and confirming the rate of biodegradability. The presence of residual soil in the fiber bundles during measurement can influence weighing results. Additionally, the decomposition and weakening of fibers, along with potential losses during removal from the soil and sample preparation for further testing, can also affect weight measurement accuracy, particularly with lignocellulosic fibers. Despite these challenges, the gravimetric method still provides valuable insight into fiber biodegradation, which should be further confirmed with other test methods.

Moreover, lignocellulosic fibers are characterized by a high degree of morphological unevenness, which remains mostly unaffected by production technology and significantly affects their properties, making it very difficult to determine the degree of biodegradation. The non-uniform geometry of lignocellulosic fibers makes property evaluation challenging, often requiring adapted test procedures and an increased number of measurements. Additionally, various factors such as cultivars, agroecological growing conditions, fiber extraction, mechanical processing, and differences in chemical composition and structure influence lignocellulosic fiber properties. The specific morphology of lignocellulosic fibers leads to the dispersion of individual measurements around the arithmetic mean when assessing fiber properties, and the multitude of parameters influencing plant growth and fiber properties complicates drawing unambiguous conclusions about the degree of biodegradation from measurement results. Therefore, determining the biodegradation rate of lignocellulosic fibers as well as other fibrous materials is an exceedingly complex process and represents a significant challenge, especially considering the absence of standardized testing methods for fiber biodegradation. The research conducted in this field is valuable for gaining knowledge and insights.

Considering all the mentioned known facts and after analyzing the obtained results, it can be concluded that the highest biodegradation occurred in hemp and jute fibers, especially in hemp fibers, likely due to their relatively lower lignin and pectin content. The other investigated fibers, namely sisal (with a higher amount of lignin and pectin), viscose (with a “skin” that slows down the microorganisms penetration), and PLA (with insufficient amount and availability of OH groups), showed lower biodegradation throughout the whole 11-day testing period.

### 3.3. Total Organic Carbon (TOC) and the Total Number of Fungi and Bacteria in the Soil

Soil organic matter (SOM) refers to the total organic material present in the soil that consists of decayed plant and animal residues, living microorganisms, soil microbe’s cells and tissues, as well as substances that soil microbes synthesize at various stages of decomposition. Total organic carbon (TOC) is a measure of the carbon stored in soil organic matter (SOM) that is used by soil microorganisms as a source of energy. When carbon-containing materials biodegrade, degrading residues migrate to the soil and cause increments in the total amount of carbon that is consumed by bacteria or fungi, causing decrements in the total amount of organic carbon in the soil. The amount of organic carbon is directly correlated with the number of microorganisms, considering that fungi and bacteria use carbon as a source of energy [42].

The influence of fibers buried for 11 days on total organic carbon (TOC) and soil microbial population (bacteria and fungi) is presented in Table 2. The soil where hemp and sisal fibers were buried had increased TOC values compared to the control soil. All the soils in which the fibers were buried had an increased number of fungi compared to the control soil. It is interesting to highlight hemp and sisal fibers again, which here recorded the smallest increase. A general increase was evident in the number of bacteria in the tested soils compared to the control soil, excluding the soil with buried hemp fibers, where a slight decrease in the bacterial population was obtained.

The higher degree of hemp fiber crystallinity (84.7%) [43], degree of polymerization (DP 2500–5500) [44], as well as the smallest degree of orientation (8°) [45] compared to jute (71%, DP 1920–4700, 8°) [46,47] and sisal fibers (75%, DP 2160, up to 23°) [47,48] reveals a greater tenacity decrease (66.2%) then sisal (17.6%) and jute fibers (24.0%). Furthermore, the number of fungi and bacteria in the soil where hemp fibers were buried (9.03 × 104, 1.25 × 107) was lower than in the soil where jute (1.03 × 105, 1.42 × 107) and sisal (9.07 × 104, 1.78 × 107) fibers were buried. Since hemp fibers have the smallest amount of lignin and pectin, which act as a protective layer, compared to jute and sisal fibers [47], it can be assumed that the decomposition process of hemp fibers started first. Respectively, fungi first attack the protective hemp layer, and then, bacteria penetrate into the fiber and degrade most of the fiber cellulose, decreasing its tenacity by 66.2%.

If jute and sisal fibers are compared, it can be observed that a higher amount of lignin and pectin protects sisal fibers, influencing the fiber degradation, which was confirmed by the obtained tenacity results; i.e., sisal fibers have a lower tenacity decrease (17.6%) compared to jute fibers (24.0%). The TOC value of soil with sisal fiber (3.08%) was higher than in the control soil (2.83%) and soil with jute fiber (2.72%). It can be assumed that fungi and bacteria attacked the protective layer, penetrating into the structure and starting to degrade the jute fiber first. The number of fungi and bacteria confirms the degradation of the jute and sisal fibers given that fungi mostly have an affinity for the protective layer and bacteria for the fibers’ cellulose molecules. The soil with buried jute fibers had a higher number of fungi and a similar bacteria number as the soil with buried sisal fibers. 

Viscose fibers are characterized by a lower degree of polymerization (DP 400–700), with a low crystallinity (40%) and degree of orientation [23,49] and a high ability to absorb moisture from the environment; thus, it was expected that fast and severe biodegradation of viscose fibers did not occur; i.e., the viscose fibers’ tenacity decrease was only 22.8%. The fiber orientation largely depends on the technological process of viscose fiber production. During production, viscose polymer consists of unoriented cellulose micelles, which in the coagulation stage are exposed to the acid bath, when the surface skin (outer layer of the fiber) of solid cellulose containing well-oriented micelles is formed. The acid then diffuses more slowly through this skin, forming an unoriented core of the viscose fibers. Besides that, stretch is a second orienting influence during viscose fiber production. During the passage of the viscose polymer through the coagulation bath, if the stretch is applied, the skin effect will increase, leaving the unoriented core of the viscose fibers. Concerning the value of the stretch and the stage at which it is applied, the viscose filaments will vary in structure from those having an oriented skin and an unoriented core, influencing its properties and thus the degradation. Considering the decrease in TOC value and increase of the fungi and bacteria population related to the control soil, it can be assumed that more time was needed for fungi and bacteria to degrade the outer layer of better-oriented molecules and to penetrate into the unoriented core of the fiber [17].

Compared to the control soil, the highest increase of fungi and bacteria was detected in the soil where PLA fibers were buried. Due to the high microorganisms increase, the TOC value of soil with PLA fibers (2.75%) decreased compared to the control soil values (2.83%). The PLA polymer degradation mechanism differs from biodegradable polymers such as hemp, jute, and sisal fibers. The PLA polymer degradation occurs in a two-step mechanism: first, chemical hydrolysis of PLA in the presence of water at elevated temperatures occurs, followed by biotic degradation by microorganisms. The microorganisms can degrade PLA only after high-molecular-weight PLA goes under hydrolysis and the molecular weight of PLA decreases. It has been reported that PLA biodegradation depends on the polymer’s degree of crystallinity, environmental conditions (temperature and humidity), and diverse microbial populations in the soil [49,50]. The combination of biodegradation conditions (temperature of 30 °C and humidity of 95%) and microbial populations in the farmland soil contributed to the chemical hydrolysis of PLA fibers, increasing the fungal and bacterial populations that degrade the fiber and decreasing the tenacity by 37.2%.

### 3.4. Morphological Analysis (SEM)

The morphological analysis and comparison of the fiber surface before and after biodegradation in the soil through scanning electron microscopy (SEM) imaging (Figure 3) revealed significant degradation caused by microorganisms after 11 days of burial. This degradation was particularly pronounced in hemp, jute, and sisal fibers and attributed to their higher moisture absorption capacity, which creates favorable conditions for microorganism development. The SEM images demonstrate that the buried hemp, jute, and sisal fibers exhibit surface damage, roughness, and prominent cracks compared to the control fibers.

The SEM images of the viscose fibers reveal surface damage, albeit with a relatively smooth appearance for the unburied fibers. Following 11 days of burial, the surface of the viscose fibers appeared smooth but exhibited numerous small defects, without any visible surface breakage. The PLA fiber displayed a smooth and clean surface before the burial test, but after 11 days of burial, the SEM images show the emergence of roughness and the presence of small pores on the PLA surface, indicating the impact of microorganism attack.

### 3.5. FTIR Analysis

The investigated fiber samples were subjected to Fourier-transform infrared spectroscopy and oscillatory spectra, which can provide insights into the changes occurring in the chemical structure of fibers after exposure to microorganisms (Figure 4, Figure 5, Figure 6, Figure 7 and Figure 8). By comparing the FTIR spectra of control and treated fibers, alterations in functional groups and chemical bonds can be identified, which can indicate the progress and extent of biodegradation.

After soil burial, the intensity of characteristic bands associated with cellulosic features such as OH stretching (around 3300 cm^−1^), CH stretching (around 2900 cm^−1^), and CH_2_ bending (around 1420 cm^−1^) showed a slight decrease. This decrease can be attributed to the breakdown and loss of cellulose chains. However, it should be noted that there were minor variations in the precise positions and intensities of these bands, influenced by factors such as the origin of the cellulose fiber, its structural characteristics, and the extraction method employed.

Regarding cellulose biodegradation, the absence of characteristic band peaks associated with hemicellulose (around 1730–1710 cm^−1^) or lignin (around 1500–1600 cm^−1^) in natural cellulose fibers (such as hemp, jute, and sisal) indicated that biodegradation primarily targeted cellulose, while hemicellulose and lignin were minimally affected (Figure 4, Figure 5 and Figure 6). However, buried hemp fibers exhibited characteristic band peaks at 1825 cm^−1^ and 1889 cm^−1^, indicating the presence of carbonyl groups (C=O) associated with carboxylic acid or anhydride functional groups. This observation suggests that the buried hemp fibers underwent biodegradation, potentially influenced by environmental factors and microbial activity [51,52]. Similar findings were also observed in buried jute fibers (peak at 1756 cm^−1^) and sisal fibers (peak at 1781 cm^−1^) [53,54].

During the process of biodegradation, microorganisms enzymatically break down the organic material, leading to changes in its chemical structure. This can result in a reduction in the intensity of characteristic bands observed in the FTIR spectra of viscose fibers (Figure 7), indicative of changes in functional groups or molecular vibrations. After biodegradation, some characteristic peaks of the viscose fibers may increase, while others remain unchanged, indicating significant alterations in the chemical structure of the fibers. A decrease in peaks at higher wavenumbers (2438 cm^−1^ and 2345 cm^−1^) signifies modifications or loss of certain functional groups, resulting in changes in molecular vibrations during biodegradation. On the other hand, an increase in peaks at lower wavenumbers (1361 cm^−1^ to 1372 cm^−1^ and 589 cm^−1^ to 614 cm^−1^) suggests the formation of new functional groups within the fiber structure [55]. These FTIR spectral changes provide evidence of the chemical transformations occurring in the viscose fibers during the biodegradation process.

Through the analysis of the control sample of PLA fibers using FTIR (Figure 8), characteristic peaks were observed. These include the peak at 3716 cm^−1^ corresponding to the stretching vibration of hydroxyl groups (O-H), the peak at 1712 cm^−1^ corresponding to the stretching vibration of the carbonyl group (C=O) in the ester linkage of PLA, the peak at 2995 cm^−1^ corresponding to the stretching vibration of C-H bonds in the methylene (CH_2_) groups of PLA, and the peak at 1155 cm^−1^ corresponding to the stretching vibration. Following exposure to microorganisms, the FTIR spectrum of PLA fibers exhibited significant changes in the peaks, with some decreasing in intensity and new peaks appearing with higher intensity. The peak at 1712 cm^−1^ increased to 1750 cm^−1^, and the peak at 1150 cm^−1^ decreased to 1129 cm^−1^, while the peaks at 3716 cm^−1^ and 2995 cm^−1^ remained unchanged, indicating substantial modifications in the chemical structure of the PLA fibers during biodegradation. The increase in the peak at 1750 cm^−1^ suggests a change in the carbonyl (C=O) stretching vibration associated with the ester linkage in PLA fibers, indicating the formation of different types of carbonyl groups during biodegradation. The decrease in the peak at 1150 cm^−1^ suggests alterations in the C-O stretching vibration in the ester linkage of PLA, indicating changes in the ester groups or the presence of different functional groups resulting from biodegradation. Additionally, no changes were observed in the peaks at 3716 cm^−1^ and 2995 cm^−1^, indicating that the hydroxyl and methylene groups remained unaffected by the biodegradation process. These observed changes in the FTIR spectrum of PLA fibers indicate modifications in the chemical structure and functional groups of the material as a result of biodegradation. These changes can be attributed to the breakdown of polymer chains and the introduction of new chemical groups during the biodegradation process. It is noteworthy that a significant number of new peaks appeared in the FTIR spectrum of PLA fibers after biodegradation, further emphasizing the substantial changes that occurred in the chemical structure of the fibers [26,56].

### 3.6. Thermogravimetric Analysis (TGA)

Thermogravimetric analysis (TGA) is commonly used to analyze the biodegradation process of natural fibers due to its ability to provide valuable insights into the thermal stability and decomposition behavior of materials. TGA analysis of the fiber’s mass loss as a function of temperature or time. During biodegradation, natural fibers undergo degradation and loss of their mass because of chemical and microbial processes. TGA allows for the quantification of mass loss, providing valuable information on the extent of biodegradation.

The characteristic temperatures obtained from TGA analysis (Table 3) after the biodegradation of cellulose fibers show its increase, which indicates an improvement in the thermal stability of the fibers after biodegradation; i.e., the fibers became more resistant to thermal decomposition. This could be attributed to changes in the chemical structure or the removal of thermally labile components during the biodegradation process. In addition, it was noticed that the remaining fiber mass after complete degradation decreased, which shows that the fibers underwent more significant degradation during the TGA analysis, which confirms the obtained results of mass and tenacity loss after biodegradation. An insignificant change in the characteristic temperatures obtained from TGA analysis of sisal fibers after biodegradation shows that the biodegradation process did not cause significant changes in the thermal stability or degradation behavior of the fibers. The sisal fiber’s thermal properties remained relatively unchanged [57].

Following biodegradation, the TGA analysis of PLA fibers revealed significant alterations in their thermal stability and decomposition behavior, indicating a notable impact of the biodegradation process on these properties. The decrease in characteristic temperature values suggests that the biodegradation process rendered the PLA fibers more susceptible to degradation at lower temperatures. Consequently, the PLA fibers underwent modifications during biodegradation that reduced their thermal stability and increased their tendency for early biodegradation. This is further supported by the observed decrease in the remaining mass after TGA analyses, affirming the efficient degradation of the PLA fibers through the biodegradation process [58].

The thermal decomposition rate can be expressed as a percentage mass loss per minute (%/min). After 11 days of the soil burial test, the thermal decomposition rate changed compared to the control samples for almost all investigated fibers. There were no significant changes in the thermal decomposition rate of jute fibers; i.e., after 11 days of the soil burial test, a slight change was recorded (from −13.926%/min to −13.911%/min), possibly influenced by factors like environmental conditions and microbial activity. The thermal decomposition rate changed by 3.254%/min (−11.382 to −14.636%/min) for hemp fibers, indicating a relatively higher mass loss due to decomposition. The thermal decomposition rate for sisal fiber ended at a very similar rate (−14.144%/min) as hemp fibers. However, compared with the control sample, the changes in the thermal decomposition rate of hemp fibers were smaller, only 0.814%/min, likely due to a higher amount of lignin. Comparing the decomposition rates, viscose fibers exhibited a much higher rate than bast fibers, likely due to their specific chemical composition and structure. Additionally, the TGA analysis of PLA fibers showed a higher thermal decomposition rate than cellulose fibers, indicating a faster degradation process, possibly influenced by the unique properties of PLA as a biodegradable polymer derived from renewable resources.

## 4. Conclusions

Based on the analysis of mass and tenacity results, it can be concluded that the influence of lignin and pectin content on the biodegradation of hemp, jute, and sisal lignocellulose fibers is more pronounced compared to their crystallinity, degree of polymerization, and fibril degree of orientation related to the long axis.

Unexpected findings were observed when analyzing the mass and tenacity loss of viscose and PLA fibers. Despite the expectation of higher biodegradability in viscose fibers due to their lower degree of polymerization, weaker orientation, and smaller proportion of crystallinity, the results showed a tenacity decrease of 22.8%, reduced total organic carbon (TOC) value, and increased populations of fungi and bacteria compared to the control soil. This suggests that the presence of a “skin” formed during the spinning process on the outer surface of the fibers hinders microorganism penetration and the biodegradation process.

In addition, PLA fibers were expected to exhibit the lowest biodegradation. However, the combination of biodegradation conditions and microbial populations in the farmland soil contributed to the chemical hydrolysis of PLA fibers, leading to an increase in fungal and bacterial populations that degraded the fiber, resulting in a tenacity decrease of 37.2%.

Morphological analysis of SEM images before and after the biodegradation process in the soil revealed significant microbial attack after 11 days of burial, particularly evident in hemp, jute, and sisal fibers. The significant changes observed in FTIR peaks and TGA analysis characteristics for each investigated fiber further confirm the findings of mass and tenacity loss, indicating the occurrence of biodegradation in the investigated fibers.

The limited availability of studies on fiber biodegradability, as revealed by the literature review, highlights the high value of the obtained data in fully understanding fiber biodegradability and facilitating the selection of suitable fibers for environmentally friendly product manufacturing.

## Figures and Tables

**Figure 1 polymers-15-03532-f001:**
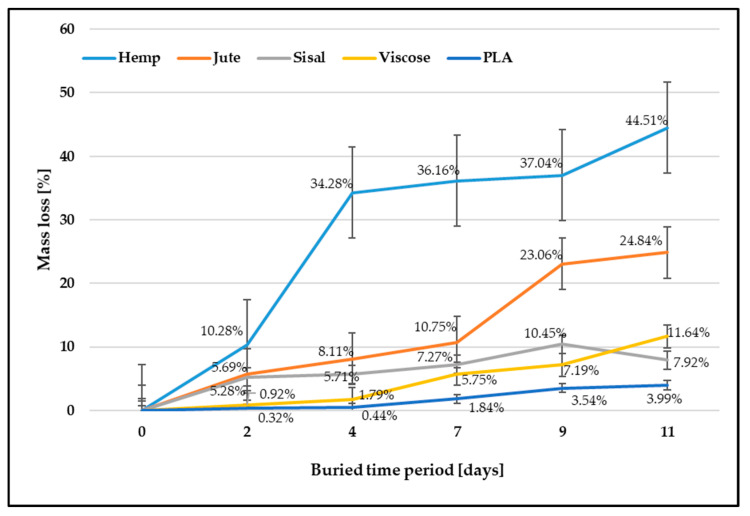
Fiber mass loss after soil burial test.

**Figure 2 polymers-15-03532-f002:**
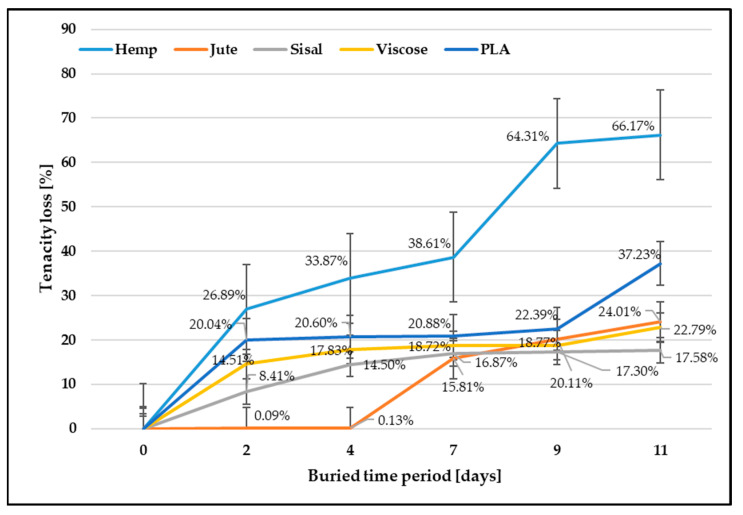
Fiber tenacity loss after soil burial test.

**Figure 3 polymers-15-03532-f003:**
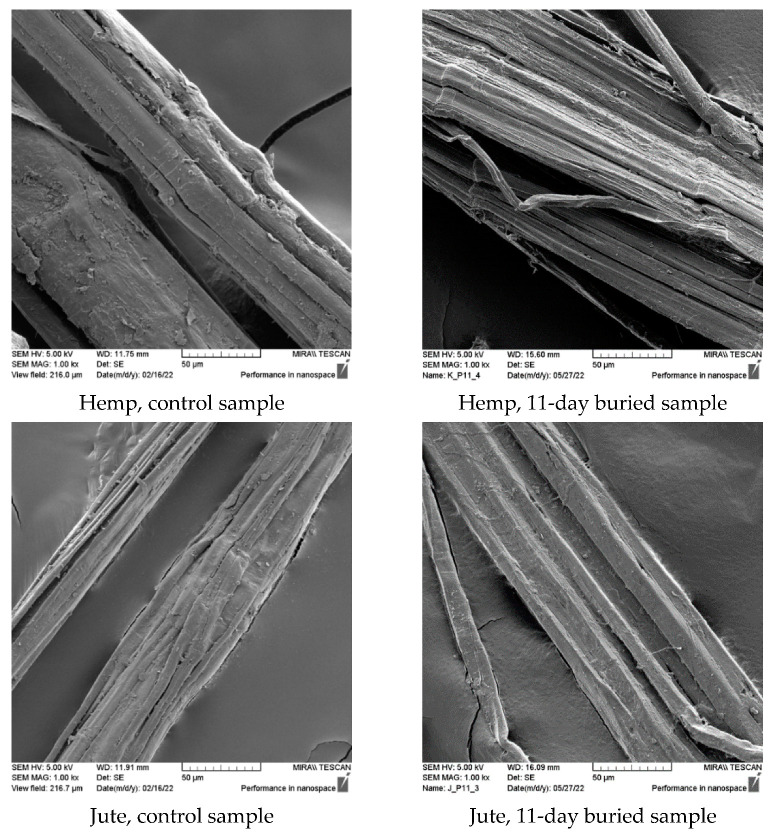
SEM images (×1000) of control and fibers after 11 days of burial.

**Figure 4 polymers-15-03532-f004:**
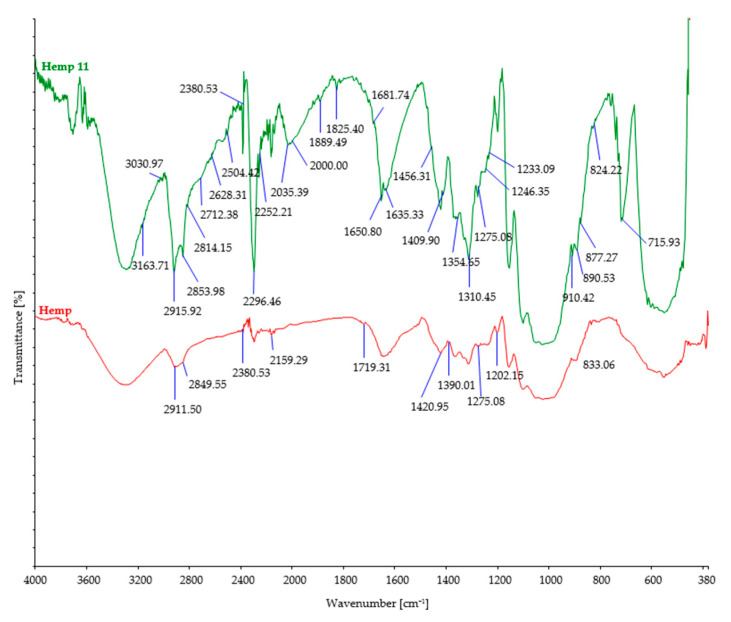
FTIR spectra of control and hemp fibers after 11 days of burial.

**Figure 5 polymers-15-03532-f005:**
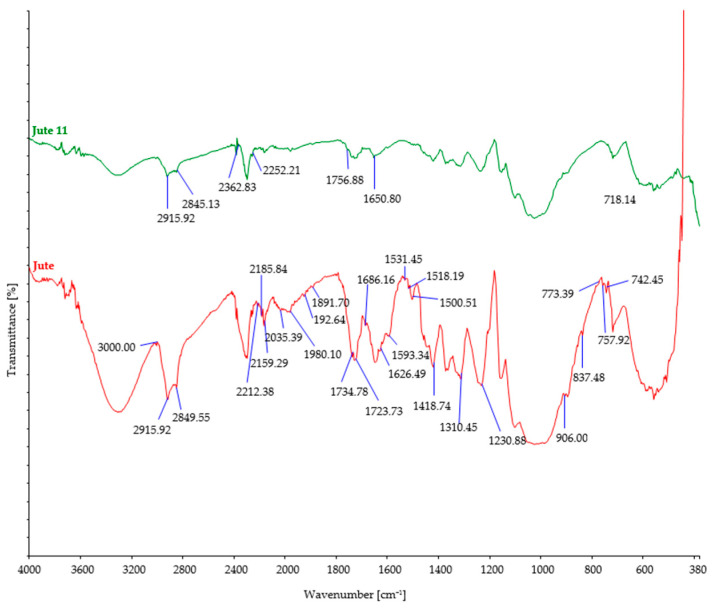
FTIR spectra of control and jute fibers after 11 days of burial.

**Figure 6 polymers-15-03532-f006:**
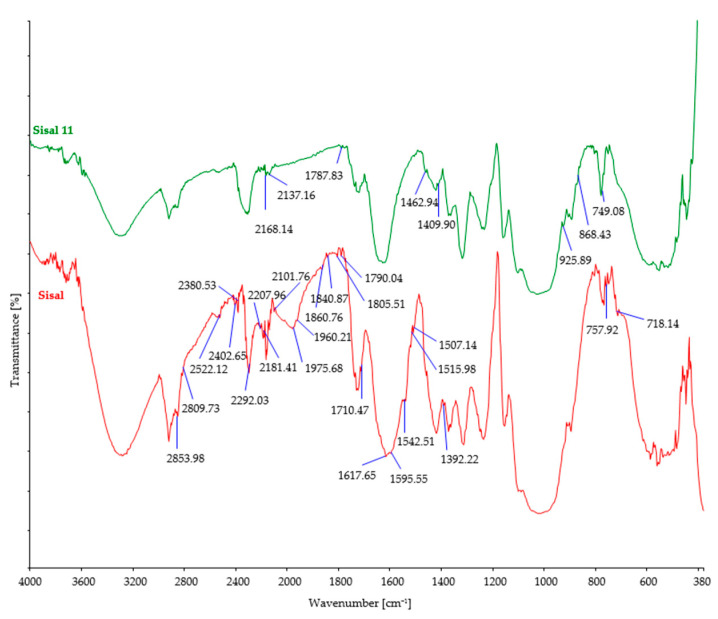
FTIR spectra of control and sisal fibers after 11 days of burial.

**Figure 7 polymers-15-03532-f007:**
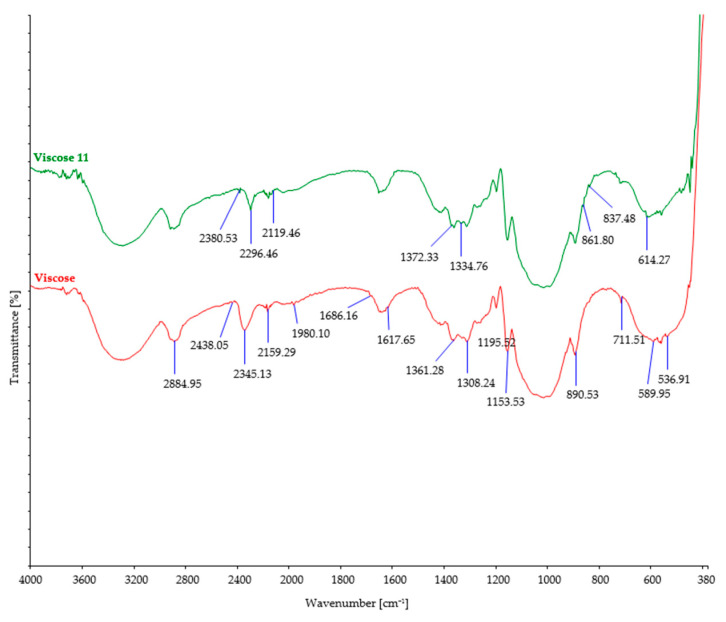
FTIR spectra of control and viscose fibers after 11 days of burial.

**Figure 8 polymers-15-03532-f008:**
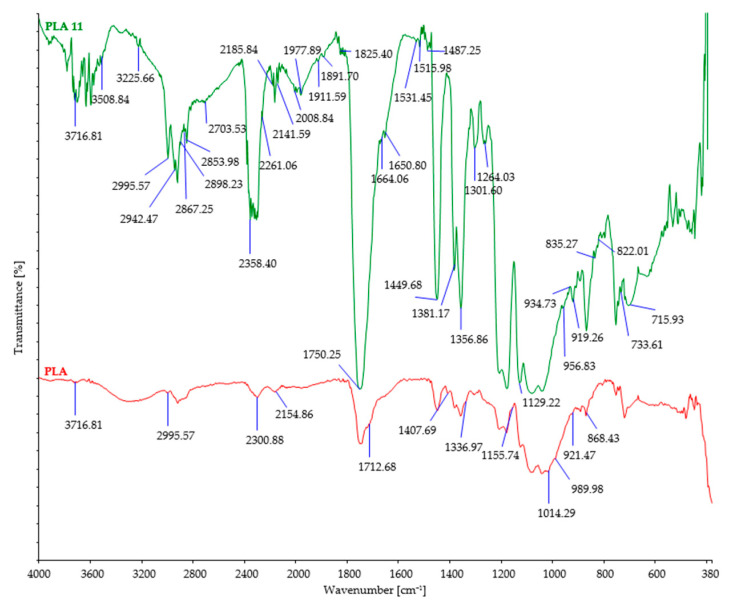
FTIR spectra of control and PLA fibers after 11 days of burial.

**Table 1 polymers-15-03532-t001:** Fiber tenacity (cN/tex) of control and buried fibers and fiber tenacity loss (%).

Samples	Control Samples	2 Days	4 Days	7 Days	9 Days	11 Days
Hemp						
$\bar{x}$, cN/tex	62.47	45.67	41.31	38.35	22.29	21.13
SD, cN/tex	21.98	13.69	16.14	13.17	7.31	6.73
CV,%	35.18	29.88	39.07	34.35	32.79	31.87
Δ (%)	/	26.89	33.87	38.61	64.31	66.17
Jute						
$\bar{x}$, cN/tex	43.94	43.90	43.88	36.99	35.10	33.39
SD, cN/tex	12.44	12.68	13.07	12.03	10.82	10.82
CV,%	28.30	28.89	29.78	32.51	30.82	32.41
Δ (%)	/	0.09	0.13	15.81	20.11	24.01
Sisal						
$\bar{x}$, cN/tex	97.88	89.64	83.66	81.36	80.94	80.67
SD, cN/tex	36.97	32.88	34.93	38.56	36.70	28.83
CV,%	37.77	36.67	41.75	47.40	45.35	35.74
Δ (%)	/	8.41	14.50	16.87	17.30	17.58
Viscose						
$\bar{x}$, cN/tex	21.36	18.26	17.55	17.36	17.35	16.49
SD, cN/tex	3.64	4.48	3.46	2.94	3.10	3.12
CV,%	10.73	24.52	19.72	16.92	17.86	18.90
Δ (%)	/	14.51	17.83	18.72	18.77	22.79
PLA						
$\bar{x}$, cN/tex	17.86	14.28	14.18	14.13	13.86	11.21
SD, cN/tex	5.36	2.29	2.50	2.95	2.90	1.53
CV,%	30.04	16.01	17.61	20.87	20.94	13.66
Δ (%)	/	20.04	20.60	20.88	22.39	37.23

Arithmetic mean,
$\bar{x}$; standard deviation, SD; coefficient of variation, CV; Δ—tenacity loss compared to the control sample.

**Table 2 polymers-15-03532-t002:** Total organic carbon (TOC) and the total number of fungi and bacteria in the soil.

Sample	TOC (%)	Fungi		Bacteria	
		$\bar{x}$ (CFU/mL)	SD (CFU/mL)	$\bar{x}$ (CFU/mL)	SD (CFU/mL)
Control soil	2.83	8.50 × 10^4^	7.55 × 10^3^	1.28 × 10^7^	7.64 × 10^5^
11-day buried sample					
Hemp	3.00	9.03 × 10^4^	3.21 × 10^3^	1.25 × 10^7^	5.03 × 10^5^
Jute	2.72	1.03 × 10^5^	3.21 × 10^3^	1.42 × 10^7^	4.36 × 10^5^
Sisal	3.08	9.07 × 10^4^	2.08 × 10^3^	1.78 × 10^7^	1.13 × 10^6^
Viscose	2.56	1.01 × 10^5^	3.51 × 10^3^	1.36 × 10^7^	5.69 × 10^5^
PLA	2.75	1.10 × 10^5^	9.85 × 10^3^	1.66 × 10^7^	1.97 × 10^6^

Arithmetic mean, $\bar{x}$; standard deviation, SD.

**Table 3 polymers-15-03532-t003:** Thermogravimetric analysis of control and fibers after 11 days of burial.

Fibers	Initial Degradation Temperature (Onset), °C	The Temperature of the Highest Decomposition Dynamics, °C	Decomposition Dynamics,%/min	Final Decomposition Temperature, °C	Residue on 850 °C,%
Hemp					
Control	318.73	358.16	−11.382	380.96	23.995
11 days buried	351.81	384.49	−14.636	410.91	13.036
Jute					
Control	368.26	379.40	−13.926	390.49	14.472
11 days buried	355.18	386.07	−13.911	410.34	14.212
Sisal					
Control	347.25	381.73	−13.330	405.62	12.793
11 days buried	347.84	382.68	−14.144	407.95	12.821
Viscose					
Control	309.85	362.54	−11.762	393.77	12.403
11 days buried	327.47	366.26	−15.692	392.14	9.805
PLA					
Control	332.99	359.26	−21.961	389.63	1.407
11 days buried	327.01	365.77	−24.359	378.05	0.742

## Data Availability

Not applicable.

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
