# Peer review of "Biodegradation Properties of Cellulose Fibers and PLA Biopolymer"

_polymers, 2023, doi:10.3390/polym15173532_

Round 1
Reviewer 1 Report (Previous Reviewer 2)
The manuscript has been improved sufficiently, my comments and questions have been answerred to satisfaction and implemented whereever possible (in one case, the authors had followed the MDPI templates, and used only initials in the author contribution section. I found that a bit cryptic but OK, if these are the rules – the comment should be adressed not to the authors but to the editioral office!).
The explanation about the ‘skin’ of viscose fibers is clear and sufficient for this purpose. Still it looks like it is cellulose in principle but has undergone fysical-mechanical changes in the spinning process. It is a too minor issue to discuss now, perhaps in follow-up research.
There is a large block of text added in the ‘results’ section, which indicates very clearly the challenges involved in assessing the biodegradability of lignocellulosic fibers. If this revised version was the first submission, I might have advised that most of this text would fit just a bit better in the introduction section – but that is such a minor comment that it is OK to leave it where it is now.
So altogether I think that this revised manuscript can now be recommended for publication in the present state.
Author Response
Thank you for your valuable feedback on the manuscript. We greatly appreciate your thorough assessment and constructive comments. We are pleased to hear that the manuscript has been improved sufficiently and that your questions and comments have been addressed to your satisfaction. We have made every effort to implement your suggestions wherever possible, and we are grateful for your attention to detail. We appreciate your support and recommendation for publication in the current state.

Reviewer 2 Report (New Reviewer)
The study analyzed biodegradation of cellulose and PLA fibers in soil, testing hemp, jute, sisal, viscose, and PLA. Results showed that lignin and pectin content impacted biodegradation of some fibers, while viscose had lower biodegradability due to a "skin" formed during spinning and PLA underwent chemical and microbial attack. The study provides insights for sustainable fiber selection and highlights a lack of research on fiber biodegradability.
Please find below my comments and suggestions:
Abstract: The length of the abstract is appropriate.
Keywords: In the keywords it is mentioned cellulose fibers but I would suggest specifying the fibers (hemp, jute,…).
Introduction. In general, the scope is presented in a clear way. In relation to the references, I would suggest the authors considering the possibility of mentioning a previously published work Biodegradation properties of natural fibers for agro textile nonwovens production - IOPscience ( https://iopscience.iop.org/article/10.1088/1757-899X/1266/1/012017)
Methodology: Materials. Although the term 'dtex' is very well-known in fibers terminology, I would recommend to mention the definition so that the paper is suitable for a broader readership.
Results and discussion.
Figure 1. I would recommend modifying the scale of the 'y-axis' so that there is not a negative scale for the mass loss (%). Usually, the units are given between brackets.
Figure 2. The same comment applies.
Table 2. A different sample code to that used in the rest of the article was used here (H11, J11, S11, CV11) whereas in the rest of the article the full name was used. This comment applies also to Table 3. In Table 3, a different sample code was used (without subscripts).
Format of the data shown in the table can also be improved.
Figure 3. For SEM images, only one fiber is observed for PLA. Is this representative of the behavior of all the PLA fibers?
Figure 4. The format of FTIR spectra should be improved
The findings and the conclusions of the research are clear, however the presentation of the results should be improved so that it is easy to read.
I would recommend the paper for publication after major revisions.
Author Response
The study analyzed biodegradation of cellulose and PLA fibers in soil, testing hemp, jute, sisal, viscose, and PLA. Results showed that lignin and pectin content impacted biodegradation of some fibers, while viscose had lower biodegradability due to a "skin" formed during spinning and PLA underwent chemical and microbial attack. The study provides insights for sustainable fiber selection and highlights a lack of research on fiber biodegradability. Please find below my comments and suggestions:
Thank you for your comments, which the authors believe will improve the manuscript
Point 1: Abstract: The length of the abstract is appropriate.
Response 1: Thank you for your comment.
Point 2: Keywords: In the keywords it is mentioned cellulose fibers but I would suggest specifying the fibers (hemp, jute,…).
Response 2: Thank you for your insightful suggestion. Adding specific fibre types (L27) to the keywords would indeed contribute to improved clarity and precision in the article's description. Your input is much appreciated!
Point 3: Introduction. In general, the scope is presented in a clear way. In relation to the references, I would suggest the authors considering the possibility of mentioning a previously published work Biodegradation properties of natural fibers for agro textile nonwovens production - IOPscience ( https://iopscience.iop.org/article/10.1088/1757-899X/1266/1/012017)
Response 3: Incorporating a previously published paper in this research would not only enhance the credibility of the study but also provide valuable context and support for the findings. It's an excellent suggestion that contributes significantly to the overall quality of the research. The authors have included the requested data at L140-148.
Point 4: Methodology: Materials. Although the term 'dtex' is very well-known in fibers terminology, I would recommend to mention the definition so that the paper is suitable for a broader readership.
Response 4: The author's opinion is that the unit "dtex" is generally known and accepted in scientific circles related to the field of textiles, and that including its definition is not necessary. However, if your stance is that it is indeed essential, the authors are willing to include it (L159-160).
Results and discussion.
Point 5: Figure 1. I would recommend modifying the scale of the 'y-axis' so that there is not a negative scale for the mass loss (%). Usually, the units are given between brackets. Figure 2. The same comment applies
Response 5: Thank you for your valuable recommendation. Authors acknowledge the need to modify the 'y-axis' scale to eliminate negative values for mass loss (%). Additionally, the units are given within brackets to enhance the clarity of the data (L266 and L315).
Point 6: Table 2. A different sample code to that used in the rest of the article was used here (H11, J11, S11, CV11) whereas in the rest of the article the full name was used. This comment applies also to Table 3. In Table 3, a different sample code was used (without subscripts). Format of the data shown in the table can also be improved.
Response 6. Thank you for your comment. Sample designations are uniform throughout the manuscript as full fibre names (L312, L380 and L514). In addition, the authors adjusted the layout and data of Table 3 (L514)
Point 7: Figure 3. For SEM images, only one fiber is observed for PLA. Is this representative of the behavior of all the PLA fibers?
Response 7: Considering that the morphological analysis of the investigated fibres has been outsourced, meaning it was conducted as an external analysis due to the current unavailability of equipment at the parent institution, authors do not have any other SEM images except for the one that has been shown in the paper. We sincerely hope that it will be accepted as such and that this deficiency will not affect the overall quality of the research.
Point 8: Figure 4. The format of FTIR spectra should be improved
Response 8: Thank you for your valuable feedback regarding the format of the FTIR spectra. The authors truly appreciate your input and understand that changing the current format at this stage may not be feasible. However, we kindly request your understanding and acceptance of the spectra in their present form for this particular stage of the research. Please rest assured that we take all comments and suggestions seriously, and we will consider your input for future revisions and improvements.
Point 9: The findings and the conclusions of the research are clear, however the presentation of the results should be improved so that it is easy to read.
Response 9: Thank you for your positive feedback on the clarity of the research findings and conclusions. We highly value your input, and we have taken your suggestion to heart to improve the presentation of the results for better readability. With great consideration for all the changes you recommended, we have worked diligently to enhance the visual and structural aspects of the results section. Our aim is to make it easier to comprehend. We sincerely hope that you find the revised presentation to be more accessible and enjoyable to read. Your valuable contribution has helped us enhance the quality of our research, and we appreciate your support and constructive feedback.
I would recommend the paper for publication after major revisions.

Round 2
Reviewer 2 Report (New Reviewer)
Dear authors,
Thank you for the revised version of the manuscript and for the highlighted sections of the document that were included.
After checking the revised version of the document, I have some comments and suggestions:
Figure 1 and 2, show a combination of a plot and the data right below in a table. The table should be labelled and included separately in case you consider that it should be included. The units are not given in the table.
The same applies for Figure 2.
In Table 1, it should be specified how fiber tenacity was determined and the units, in case you use fiber tenacity or if the measurements referred to reduced tenacity. Please clarify.
In Table 2, the data in the table should be displayed using scientific notation instead of using the data written as "9.03E+04", for instance.
The presentation of FTIR spectra should be improved. The units of the axis cannot be read. It should say Transmitance (%) vs wavenumber (cm-1).
These are the most critical aspects, but I recommend a thoroughout revision of the document.
Author Response
Point 1: Figure 1 and 2, show a combination of a plot and the data right below in a table. The table should be labelled and included separately in case you consider that it should be included. The units are not given in the table.
The same applies for Figure 2.
Response 1: We extend our gratitude for your valuable recommendation, which has contributed to the improvement of our manuscript. Additionally, the units are given within brackets, at lines (not as a table) to enhance the clarity of the data (L270 and L319).
Point 2: In Table 1, it should be specified how fiber tenacity was determined and the units, in case you use fiber tenacity or if the measurements referred to reduced tenacity. Please clarify.
Response 2: Thank you for your insightful suggestion, which has enhanced the clarity of Table 1. The authors change the title of Table 1 at L315
Point 3: In Table 2, the data in the table should be displayed using scientific notation instead of using the data written as "9.03E+04", for instance.
Response 3: Thank you for your suggestion. The authors recognize the necessity of employing scientific notation in Table 2 (L383), adhering to the conventions of the International System of Units (SI).
Point 4: The presentation of FTIR spectra should be improved. The units of the axis cannot be read. It should say Transmittance (%) vs wavenumber (cm-1).
Response 4: Thank you for your valuable feedback regarding the format of the FTIR spectra. The authors incorporated the suggestion which significantly enhanced the clarity of the FTIR spectra (L470, L473, L475, L491 and L519).
These are the most critical aspects, but I recommend a thoroughout revision of the document.
We sincerely thank you for your valuable comments and suggestions on the manuscript. Your thorough assessment and constructive feedback have significantly enhanced the quality of the manuscript.

This manuscript is a resubmission of an earlier submission. The following is a list of the peer review reports and author responses from that submission.
Round 1
Reviewer 1 Report
This study reports the biodegradable characteristics of PLA composites with different cellulose fibers. This present manuscript is needed to be carefully revised in order to publish in Polymers.
Here are some main concerns:
1. In the methodology section, there was no information regarding fiber extraction and preparation. Moreover, no details of the fabrication of cellulose-PLA composites were found.
2. The mass losses reported in Figure 1 and the tenacity shown in Figure 2 must contain error bars.
3. The characterization of the materials (e.g., spectroscopic investigations, microscopic studies, and thermal property examinations) before and after the burying process must be given to enrich the results and discussion section.
4. The degradation mechanism must be clearly given. The related schematic diagram and/or the chemical reactions involved must be included to provide a better understanding of the degradation characteristics of the prepared materials.
5. Most of the references are out-of-date. There was only one reference in 2021. The rest was older.
6. The conclusion section is needed to be more concise.
Moderate editing of English language required
Author Response
Thank you for your comments, which the authors believe will improve our manuscript

Reviewer 2 Report
This article is about biodegradation of several, mostly cellulose containing fibers. The purpose of the research is well explained as is the background and the challenges involved. There are quite interesting results obtained, not always as-expected but that makes research always challenging. The mechanisms of biodegradation are discussed, and together with the results obtained I think that this manuscript can be a valuable contribution to increased understanding of this important and timely topic.
The English is of generally good quality, although a grammar check would be advisable since some sentences have an illogic sequence. Such does not really affect understanding the statements, but it can be made to look better. A few sentences are too long and had better be split up for easier reading. I will indicate two cases in the more detailed comments. BTW, the hyphening of the title looks strange (bi-opolymer).
I do have some comments and questions that need to be answered and some revisions accordingly are recommended.
It is important that the reader has easy access to the results as presented in graphs and tables. Therefore is must be recommended that full names of the materials teste are mentioned in the axis legends. Now it is a puzzle where to find the meaning of the abbreviations.
The list of author contributions had better be made with full names, preferably not initials that require the reader to solve a kind of cryptogram..
Detailed comments are in the section 'to the authors'.
I think that just a bit more than minor revisions are needed, to I select the recommendation 'major'. It would be nice to also have an option 'moderate revisions'!
DETAILED COMMENTS-QUESTIONS TO THE AUTHORS
INTRO
The sentence starting at L 27 is quite complex. Probably a split up in smaller statements would make this information more accessible. The same at L 81.
I see a certain contradiction in this statements: 'abiotic biodegradation is a reaction that leads to molecules that are.. biodegradable. Perhaps the first sentence of this paragraph is too simple, since biodegradation is not a single chemical reaction but a collective noun for a complex and interacting set of chemical, biochemical and physical processes.
L 43 paragraph. This statement really appeals to me since I think there is a pitfall in just doing gravimetry, with its associated uncertainties e.g. in the separation of the residue of the original materials from the surrounding soil matrix.
METHODOLOGY
Here, standardized methods are only indicated by ISO-numbers. I can not see if the further explanation complies to these methods or that the authors have used adaptations. This is seen several times in this paragraph. Please make this a bit more clear.
RESULTS
These are mostly clear, and the explanation of behavior that was not expected makes sense. A few comments/questions:
L 194 Is it correct to say that a degraded piece of any of the polymers is returned to monomer? I think not - a monomer can be polymerized again, but I presume that these are individual molecules of a kind of sugar (glucose or anhydro-)?
Figure 1: this is an example of where abbreviations had better be replaced by full names, in the graph or in the legend. Now it is puzzle where to find what the letters mean. This is much better represented in Table 1! BTW, Fig. 2 is also called Fig.1 which is presumably just a typing error. But the same comment on the letters applies here. There are also three paragraphs 3.1 !
L 243 It would be interesting to know what properties and composition can be expected from what is called 'the skin'. If it cellulose as well, what has happened to the molecules that makes it so much more resistant to breakdown?
L 309 two times 'expected' looks strange.
CONCLUSIONS
OK, no comments
This article is about biodegradation of several, mostly cellulose containing fibers. The purpose of the research is well explained as is the background and the challenges involved. There are quite interesting results obtained, not always as-expected but that makes research always challenging. The mechanisms of biodegradation are discussed, and together with the results obtained I think that this manuscript can be a valuable contribution to increased understanding of this important and timely topic.
The English is of generally good quality, although a grammar check would be advisable since some sentences have an illogic sequence. Such does not really affect understanding the statements, but it can be made to look better. A few sentences are too long and had better be split up for easier reading. I will indicate two cases in the more detailed comments. BTW, the hyphening of the title looks strange (bi-opolymer).
I do have some comments and questions that need to be answered and some revisions accordingly are recommended.
It is important that the reader has easy access to the results as presented in graphs and tables. Therefore is must be recommended that full names of the materials teste are mentioned in the axis legends. Now it is a puzzle where to find the meaning of the abbreviations.
The list of author contributions had better be made with full names, preferably not initials that require the reader to solve a kind of cryptogram..
Detailed comments are in the section 'to the authors'.
I think that just a bit more than minor revisions are needed, to I select the recommendation 'major'.
DETAILED COMMENTS-QUESTIONS TO THE AUTHORS
INTRO
The sentence starting at L 27 is quite complex. Probably a split up in smaller statements would make this information more accessible. The same at L 81.
I see a certain contradiction in this statements: 'abiotic biodegradation is a reaction that leads to molecules that are.. biodegradable. Perhaps the first sentence of this paragraph is too simple, since biodegradation is not a single chemical reaction but a collective noun for a complex and interacting set of chemical, biochemical and physical processes.
L 43 paragraph. This statement really appeals to me since I think there is a pitfall in just doing gravimetry, with its associated uncertainties e.g. in the separation of the residue of the original materials from the surrounding soil matrix.
METHODOLOGY
Here, standardized methods are only indicated by ISO-numbers. I can not see if the further explanation complies to these methods or that the authors have used adaptations. This is seen several times in this paragraph. Please make this a bit more clear.
RESULTS
These are mostly clear, and the explanation of behavior that was not expected makes sense. A few comments/questions:
L 194 Is it correct to say that a degraded piece of any of the polymers is returned to monomer? I think not - a monomer can be polymerized again, but I presume that these are individual molecules of a kind of sugar (glucose or anhydro-)?
Figure 1: this is an example of where abbreviations had better be replaced by full names, in the graph or in the legend. Now it is puzzle where to find what the letters mean. This is much better represented in Table 1! BTW, Fig. 2 is also called Fig.1 which is presumably just a typing error. But the same comment on the letters applies here.
There are also three paragraphs 3.1 !
L 243 It would be interesting to know what properties and composition can be expected from what is called 'the skin'. If it cellulose as well, what has happened to the molecules that makes it so much more resistant to breakdown?
L 309 two times 'expected' looks strange.
CONCLUSIONS
OK, no comments
Author Response

(The authors gave the same response as above.)

Round 2
Reviewer 1 Report
The concerns have been addressed well so the revised manuscript is recommended to publish as it is.
Author Response

(The authors gave the same response as above.)
